# On Softmax Direct Preference Optimization for Recommendation

**Yuxin Chen**[1*] **Junfei Tan**[2*] **An Zhang**[1†]
**Zhengyi Yang**[2] **Leheng Sheng**[1] **Enzhi Zhang**[3]
**Xiang Wang**[2] **Tat-Seng Chua**[1]
[1]National University of Singapore
[2]University of Science and Technology of China
[3]Hokkaido University
yuxin.chen@u.nus.edu, sober_clever@mail.ustc.edu.cn, anzhang@u.nus.edu
leheng.sheng@u.nus.edu, enzhi.zhang.n6@elms.hokudai.ac.jp
yangzhy1998@gmail.com, xiangwang1223@gmail.com, dcscts@nus.edu.sg

## Abstract

Recommender systems aim to predict personalized rankings based on user preference data. With the rise of Language Models (LMs), LM-based recommenders have been widely explored due to their extensive world knowledge and powerful reasoning abilities. Most of the LM-based recommenders convert historical interactions into language prompts, pairing with a positive item as the target response and fine-tuning LM with a language modeling loss. However, the current objective fails to fully leverage preference data and is not optimized for personalized ranking tasks, which hinders the performance of LM-based recommenders. Inspired by the current advancement of Direct Preference Optimization (DPO) in human preference alignment and the success of softmax loss in recommendations, we propose Softmax-DPO (**S-DPO**) to instill ranking information into the LM to help LM-based recommenders distinguish preferred items from negatives, rather than solely focusing on positives. Specifically, we incorporate multiple negatives in user preference data and devise an alternative version of DPO loss tailored for LM-based recommenders, which is extended from the traditional full-ranking Plackett-Luce (PL) model to partial rankings and connected to softmax sampling strategies. Theoretically, we bridge S-DPO with the softmax loss over negative sampling and find that it has an inherent benefit of mining hard negatives, which assures its exceptional capabilities in recommendation tasks. Empirically, extensive experiments conducted on three real-world datasets demonstrate the superiority of S-DPO to effectively model user preference and further boost recommendation performance while providing better rewards for preferred items. Our codes are available at https://github.com/chenyuxin1999/S-DPO.

## 1 Introduction

Recommender systems aim to predict personalized rankings based on user preference data, *i.e.,* historical interactions such as purchases, clicks, and ratings [1, 2]. Recently, leveraging the extensive world knowledge and powerful reasoning abilities of language models (LMs) [3–6], LM-based recommenders have been broadly explored [7–9]. These recommenders convert historical interaction data into language prompts and either perform in-context learning or fine-tune LMs, demonstrating notable

---

[*]These authors contributed equally to this work.

[†]An Zhang is the corresponding author.

38th Conference on Neural Information Processing Systems (NeurIPS 2024).

advantages, including zero-shot and few-shot reasoning [10–13], enhanced generalization abilities [14, 15], and rich semantic understanding [16–19]. However, current LM-based recommenders typically utilize language modeling loss for personalized ranking objectives—predicting the next token—which significantly differs from the objective of modeling user preferences in recommendation tasks [20, 21].

We argue that the current objective of LM-based recommenders does not fully utilize preference data and is not optimized for personalized ranking tasks, thereby hindering recommendation performance. Most LM-based recommenders address recommendation tasks by leveraging specialized language prompts [14, 17, 22–24], incorporating collaborative signals as a new modality [18, 19, 25], or extending the vocabulary of LMs with item tokens [16, 26–30]. Typically, these recommenders pair each language prompt, including the user's historical interaction item lists, with a single positive item and then update LM parameters using language modeling loss [14, 19]. Despite being designed for recommendation tasks, these LM-based recommenders do not consider negative items and are not directly optimized for personalized rankings. Such a training paradigm fails to fully leverage user preference data and overlooks the role of negative items in recommendations, thereby impeding the alignment of LMs with user preferences.

Inspired by the success of using human-labeled data to align LMs with human preferences [31–33] and advancements in direct preference optimization (DPO) [34–36], we make progress on aligning LMs with recommendations by fine-tuning them to predict the next item in accordance with the user's preference. This preference alignment stage aims to instill ranking information into the LMs and help recommenders distinguish preferred items from negatives, rather than solely focus on positives.

Towards this end, we incorporate multiple negatives in user preference data and devise an alternative version of DPO loss tailored for recommendation, connected to softmax sampling strategies [20, 37–39], which we call **S-DPO**. Specifically, we first devise supervised fine-tuning to inject domain knowledge and improve LM's ability to follow the instructions before preference alignment phase, following [14, 33]. In the preference alignment stage, instead of constructing solely positive pairs, we initially pair each language prompt with both positive and randomly sampled multiple negatives to build text-based preference data. Building upon these preference data, we extend conventional DPO with the Bradley-Terry preference model [34, 40] on pairwise data to the Plackett-Luce preference model [41, 42], which handles relative rankings among multiple samples. Furthermore, we generalize the traditional Plackett-Luce preference model, which is designed for full relative rankings, to accommodate partial rankings, a more natural fit for recommendation tasks.

Benefiting from the multiple negatives in preference data, S-DPO offers three appealing properties. On the one hand, S-DPO serves as the first specialized personalized ranking loss for LM-based recommenders, effectively utilizing multiple negatives and acknowledging the importance of preference data. Empirically, we demonstrate that it provides more effective ranking gradients and better rewards for preferred items compared with DPO (*cf.* Section 4.2). On the other hand, we theoretically bridge the DPO loss with the pairwise BPR loss [43] over pairwise data and connect S-DPO with the softmax loss over negative sampling (also known as contrastive loss in self-supervised recommendations, which achieves state-of-the-art performance [37, 44, 45]). This connection naturally underscores the ranking performance of S-DPO and highlights the critical role of multiple negatives. Furthermore, gradient analysis demonstrates that S-DPO has an inherent benefit of mining hard negative examples similar to contrastive learning paradigm [38], which not only boosts the performance but also accelerates the training process (*cf.* Section 3.1), assuring its exceptional capabilities in recommendation tasks.

Overall, our contributions can be concluded as follows:

- We are among the first to point out that the widely used language modeling loss in LM-based recommendation is not designed for ranking tasks and fails to fully utilize user preference data, thereby hindering recommendation performance.

- We propose S-DPO, an alternative version of DPO loss extended from the traditional Plackett-Luce preference model, incorporating multiple negatives to instill ranking information into LM and tailoring for LM-based recommenders.

- We theoretically bridge S-DPO with the softmax loss over negative sampling to highlight the critical role of multiple negatives and find its inherent benefit of mining hard negatives, assuring its capabilities.

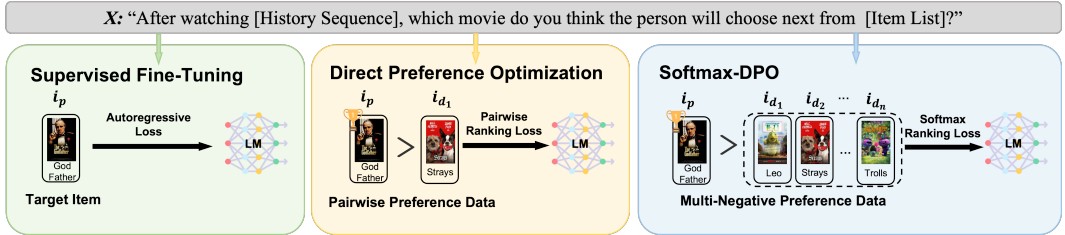

Figure 1: Framework of S-DPO. Different from existing methods which fine-tune LMs with a language modeling loss without tailoring for recommendations, S-DPO proposes to explicitly instill ranking information into LMs. To take one step further, S-DPO incorporates multiple negatives in user preference data and generalizes pairwise DPO loss to softmax ranking loss.

## 2   Preliminary

In this section, we first formalize sequential recommendation as the task of aligning language models (LMs) with user preferences. Then, we discuss the general framework of current LM-based recommenders that utilizes language modeling loss to fine-tune LMs. Finally, we outline the training process widely used to align LMs with human preferences, including reinforcement learning from human feedback (RLHF) and direct preference optimization (DPO).

**Task Formulation.**   Given the historical interactions $\mathcal{H}_u$ of one user $u$ in chronological order, the goal of LM-based sequential recommender $\mathcal{M}_\theta$, where $\theta$ represents trainable parameters, is to select the item $i_p$ preferred by user $u$ from candidate set $C = \{i_j\}_{j=1}^N$, where $N$ is the number of candidates. This task requires that item $i_p$ be preferred over the other candidate items, denoted by $\mathcal{I}_d = C \backslash \{i_p\}$. This requirement explicitly defines a multi-negative preference understanding for LM-based recommenders, which can be formulated as follows:

$$\forall i_d \in \mathcal{I}_d, \quad i_p \succ_u i_d, \tag{1}$$

wherein $\succ_u$ stands for the preference of user $u$.

**Fine-tuning LM-based recommenders.**   Current LM-based recommenders widely adopt supervised fine-tuning (SFT) [33] on recommendation-specific data to enhance their performance [7, 9]. Generally, this involves two steps: structuring recommendation data as text-based pairs and then fine-tuning LMs based on these pairs. In the first step, for user $u$, a recommendation task prompt $x_u$ encompasses the user's historical interactions $\mathcal{H}_u$, the candidate item set $C$, and a description of the sequential recommendation task. This prompt $x_u$ is paired with the title of the preferred item $i_p$ in the candidate set $C$, denoted as $e_p$, to form the pair data $(x_u, e_p)$. In the second step, the $(x_u, e_p)$ pairs are utilized to fine-tune the LM-based recommender $\mathcal{M}_\theta$ through language modeling loss. This loss, commonly used in SFT in language modeling tasks, implicitly treats the recommendation task as predicting the next token based on preceding tokens. Formally, the objective of optimizing the LM-based recommender $\mathcal{M}_\theta$ with pair data $(x_u, e_p)$ can be formulated as:

$$\max_\theta \sum_{(x_u, e_p)} \sum_{t=1}^{|e_p|} \log(P_\theta((e_p)_t | x_u, (e_p)_{<t}), \tag{2}$$

where $|e_p|$ is the number of tokens in $e_p$, $(e_p)_t$ is the $t$-th token of $e_p$ and $(e_p)_{<t}$ is the tokens preceding $(e_p)_t$.

However, recommendation tasks are essentially user preference alignment tasks, as formalized in the above task formulation, and differ from language modeling tasks that consider only positive responses. Such a gap necessitates further exploration into aligning LM-based recommenders with user preference, an area that has been underexplored.

**RLHF pipeline and DPO.**   Recent studies in natural language processing (NLP) explore the use of human-labeled pairwise data as a reward signal to align LMs with human preferences, such as RLHF [33] and DPO [34]. Specifically, the RLHF [33] pipeline adds two additional phases after the SFT phase: reward model training and reinforcement learning (RL) optimization. After obtaining the SFT model $\pi^{\text{SFT}}$, RLHF further optimizes it with pairwise preference data.

Inspired by the success of RLHF in NLP, we leverage RLHF to inject recommendation-specific user pairwise preference into LM-based recommenders. Let $\mathcal{E} = \{e_j\}_{j=1}^{N}$ denote the title set of candidate items, where $e_j$ denotes the title of item $i_j$. Given two items $i_j, i_k \in \mathcal{C}$, the user preference $i_j >_u i_k$ can be seamlessly transformed into a response preference, stipulating that $e_j$ is preferred over $e_k$ when presented with prompt $x_u$, denoted as $e_j \succ e_k | x_u$. By sampling one dispreferred item $i_d$ from dispreferred candidate set $\mathcal{I}_d$, we can curate a preference dataset $\{(e_p, e_d, x_u)\}$.

After that, RLHF utilizes a preference model for preference distribution modeling, such as Bradley-Terry (BT) model [40]. This preference model assumes there is a latent function $r(x_u, e_j)$ representing the reward of prompt-response pair $(x_u, e_j)$. The bigger reward $r(x_u, e_j)$ means the more user $u$ prefers item $i$. From this perspective, reward function $r(x_u, e_j)$ serves as a scoring function that quantifies the preference of user $u$ to item $i$. Besides, the preference model defines a mapping from the reward function $r(x_u, e_j)$ to a preference distribution $p_r(e_j \succ e_k | x_u)$. Based on preference distribution, an optimal reward function is trained by maximizing the likelihood of preference data. The training objective of this phase is as follows:

$$\mathcal{L}_{\mathrm{RM}} = -\mathbb{E}_{(x_u, e_p, e_d)}[\log p_r(e_p \succ e_d | x_u)]. \tag{3}$$

Let $\pi_\theta(e|x_u)$ be the probability that LM-based recommender $\mathcal{M}_\theta$ output title $e$ given prompt $x_u$. The final reinforcement learning phase aims to maximize the expected reward of policy while not deviate too far from the reference model, formulating the following objective for optimal policy:

$$\max_{\pi_\theta} \mathbb{E}_{x_u \sim \mathcal{D}, e \sim \pi_\theta(e|x_u)}[r(x_u, e)] - \beta \mathbb{D}_{\mathrm{KL}}[\pi_\theta(e|x_u)||\pi_{\mathrm{ref}}(e|x_u)], \tag{4}$$

where $\mathcal{D}$ denotes the distribution of $x_u$ and $\pi_{\mathrm{ref}} = \pi^{\mathrm{SFT}}$.

A recent study, DPO [34], theoretically proves the optimal policy in a closed form to Eq.(4) is

$$\pi^*(e|x_u) = \frac{1}{Z(x_u)} \pi_{\mathrm{ref}}(e|x_u) \exp\left(\frac{1}{\beta} r(x_u, e)\right), \tag{5}$$

which is equivalent to

$$r(x_u, e) = \beta \log \frac{\pi(e|x_u)}{\pi_{\mathrm{ref}}(e|x_u)} + \beta \log Z(x_u), \tag{6}$$

where $Z(x_u) = \sum_e \pi_{\mathrm{ref}}(e|x_u) \exp\left(\frac{1}{\beta} r(x_u, e)\right)$ is the partition function.

By defining $p_r(e_p \succ e_d | x_u)$ as $\sigma(r(x_u, e_p) - r(x_u, e_d))$ in Eq.(3) according to the BT model used in RLHF and substituting term $r(x_u, e)$ in Eq.(3) with Eq.(6), the last two phases of RLHF pipeline can be equivalently transformed into optimizing DPO loss below:

$$\mathcal{L}_{\mathrm{DPO}} = -\mathbb{E}_{(x_u, e_p, e_d)}\left[\log \sigma\left(\beta \log \frac{\pi_\theta(e_p|x_u)}{\pi_{\mathrm{ref}}(e_p|x_u)} - \beta \log \frac{\pi_\theta(e_d|x_u)}{\pi_{\mathrm{ref}}(e_d|x_u)}\right)\right], \tag{7}$$

wherein $\sigma(x)$ is the sigmoid function.

DPO is able to directly extract the optimal policy from pairwise preference data, making it more practical for preference alignment than RLHF. Nevertheless, DPO and RLHF are usually designed for pairwise preference. The oversight of other negative items impedes the performance of the LM-based recommenders. To bridge the gap, we expand DPO to S-DPO in recommendation tasks, in consideration of multiple negative items.

## 3 Methodology

### 3.1 Derivation of S-DPO loss

To align LM-based recommender $\mathcal{M}_\theta$ with multi-negative preference, we first derive the preference distribution and then propose a new loss function called S-DPO (depicted in Figure 1).

**Multi-negative Preference Distribution.** As mentioned in Section 2, for user $u$, there is a partial ranking stipulating $i_p \succ_u i_d, \forall i_d \in \mathcal{I}_d$ in sequential recommendation tasks. Let $\mathcal{E}_d$ be the titles of dispreferred items $\mathcal{I}_d$. The aforementioned partial ranking is equivalent to $e_p \succ e_d | x_u, \forall e_d \in \mathcal{E}_d$, from which a multi-negative preference dataset $\{x_u, e_p, \mathcal{E}_d\}$ can be curated in an analogous way to RLHF.

For the dataset pairing one preferred item with multiple dispreferred items, we leverage the Plackett-Luce (PL) model [41, 42] to build preference distribution. Given prompt $x_u$, $K$ titles $e_1, e_2, \cdots, e_K$ and a permutation $\tau : [K] \rightarrow [K]$ reflecting the user preference, with $\tau(j)$ denoting the $j$-th element of permutation $\tau$, the PL model estimates that the ranking $e_{\tau(1)}, e_{\tau(2)}, \cdots, e_{\tau(K)}$ turns out true, as:

$$p(\tau|e_1, e_2, \cdots, e_K, x_u) = \prod_{j=1}^{K} \frac{\exp\left(r(x_u, e_{\tau(j)})\right)}{\Sigma_{l=j}^{K}\exp(r\left(x_u, e_{\tau(l)}\right))}. \tag{8}$$

By enumerating all the permutations starting with $p$ and calculating sum of their probability given by the PL model, the final multi-negative preference distribution $p^*$ can be derived as:

$$p^*(e_p \succ e_d, \forall e_d \in \mathcal{E}_d | x_u) = \frac{\exp(r(x_u, e_p))}{\sum_{j=1}^{K} \exp(r(x_u, e_j))}. \tag{9}$$

For brevity, the complete derivation is delegated to Appendix A.1.

**Deriving S-DPO.** By substituting reward function $r(x_u, e)$ in Eq.(9) with Eq.(6), the multi-negative preference distribution can be rewritten as:

$$p^*(e_p \succ e_d, \forall e_d \in \mathcal{E}_d | x_u) = \frac{1}{1 + \sum_{e_d \in \mathcal{E}_d} \exp\left(\beta \log \frac{\pi(e_d|x_u)}{\pi_{\text{ref}}(e_d|x_u)} - \beta \log \frac{\pi(e_p|x_u)}{\pi_{\text{ref}}(e_p|x_u)}\right)}. \tag{10}$$

Through plugging distribution given by Eq.(10) in the reward learning objective in Eq.(3), our S-DPO loss can be formulated for policy $\pi_\theta$ as:

$$\mathcal{L}_{\text{S-DPO}}(\pi_\theta; \pi_{\text{ref}}) = -\mathbb{E}_{(x_u, e_p, \mathcal{E}_d) \sim \mathcal{D}} \left[ \log \sigma \left( -\log \sum_{e_d \in \mathcal{E}_d} \exp\left( \beta \log \frac{\pi_\theta(e_d|x_u)}{\pi_{\text{ref}}(e_d|x_u)} - \beta \log \frac{\pi_\theta(e_p|x_u)}{\pi_{\text{ref}}(e_p|x_u)} \right) \right) \right]. \tag{11}$$

Notably, when the number of candidates $N$ is 2, which means there is only one dispreferred item, S-DPO reduces to DPO. The proof is provided in Appendix A.2.

**Gradient Analysis.** We conduct gradient analysis on S-DPO. The gradient of $\mathcal{L}_{\text{S-DPO}}$ with respect to parameters $\theta$ takes the following formulation:

$$\nabla_\theta \mathcal{L}_{\text{S-DPO}}(\pi_\theta; \pi_{\text{ref}}) =$$

$$- \beta \mathbb{E}_{(x_u, e_p, \mathcal{E}_d)} \Bigg[ \underbrace{\sigma \left( \log \sum_{e_d \in \mathcal{E}_d} \exp(g(e_d, e_p, x_u)) \right)}_{\text{higher weight when reward deviates from preference}} \cdot \Bigg[ \nabla_\theta \log \pi_\theta(e_p|x_u) - \sum_{e_d \in \mathcal{E}_d} \underbrace{\frac{\nabla_\theta \log \pi_\theta(e_d|x_u)}{\sum_{e'_d \in \mathcal{E}_d} \exp(g(e'_d, e_d, x_u))}}_{\text{higher weight when reward is larger}} \Bigg] \Bigg],$$

wherein $g(e_j, e_k, x_u) = r_\theta(x_u, e_j) - r_\theta(x_u, e_k)$ and similar to DPO, $r_\theta(x_u, e) = \beta \log \frac{\pi_\theta(e|x_u)}{\pi_{\text{ref}}(e|x_u)}$ is the implicit reward function defined by $\pi_\theta$. See Appendix A.3 for a complete derivation.

Recap the DPO gradient below:

$$\nabla_\theta \mathcal{L}_{\text{DPO}}(\pi_\theta; \pi_{\text{ref}}) = -\beta \mathbb{E}_{(x_u, e_p, e_d)} \Bigg[ \underbrace{\sigma(g(e_d, e_p, x_u))}_{\text{higher weight when reward is wrong}} \cdot [\nabla_\theta \log \pi_\theta(e_p|x_u) - \nabla_\theta \log \pi_\theta(e_d|x_u)] \Bigg].$$

Similar to DPO, the gradient of S-DPO loss increases the likelihood of the preferred item and decreases the likelihood of all the dispreferred items. Each example is also weighed by how much the

implicit reward $r(x_u, e)$ deviates from the preference data. However, compared with DPO, S-DPO harnesses information of multiple dispreferred items in this weight.

Moreover, S-DPO treats gradients of different negative (dispreferred) items differently by assigning the gradient of each negative item with an extra weight $\frac{1}{\sum_{\epsilon'_d \in \mathcal{E}_d} \exp(g(e'_d, e_d, x_u))} = \frac{\exp(r_\theta(x_u, e_d))}{\sum_{\epsilon'_d \in \mathcal{E}_d} \exp(r_\theta(x_u, e'_d))}$. This term reflects the relative reward of each negative item compared with other negative items. Similar to [38], we can categorize negative items into two categories: (1) Hard negative items, whose reward $r_\theta(x_u, e_d) = \beta \frac{\pi_\theta(e_d|x_u)}{\pi_{\text{ref}}(e_d|x_u)}$ is relatively high, making it more probable to be chosen by LM-based recommenders; (2) Easy negative items, whose reward $r_\theta(x_u, e_d)$ is relatively low, making it less likely to be output. For hard negative items, the extra weight term $\frac{\exp(r_\theta(x_u, e_d))}{\sum_{\epsilon'_d \in \mathcal{E}_d} \exp(r_\theta(x_u, e'_d))}$ tends to be larger, leading to more decline for likelihood. This mechanism makes LM-based recommenders more discriminative and endows S-DPO with more effectiveness and stability than DPO.

## 3.2 Properties of S-DPO

In this section, we will discuss the structural correlation between DPO and BPR [43], together with S-DPO and softmax loss [39], which demonstrates the advantage of S-DPO over DPO and language modeling loss.

For user $u$, preferred item $i_p$ and one dispreferred $i_d \in \mathcal{I}_d$, BPR loss takes the form:

$$\mathcal{L}_{\text{BPR}} = -\mathbb{E}_{(u,i_p,i_d)} \left[ \log \sigma \left( f(u, i_p) - f(u, i_d) \right) \right], \tag{12}$$

wherein $f(u, i)$ represents preference score of user $u$ for item $i$.

Similarly, given dispreferred item set $\mathcal{I}_d$, the softmax loss takes the form:

$$\mathcal{L}_{\text{softmax}} = -\mathbb{E}_{(u,i_p,\mathcal{I}_d)} \left[ \log \sigma \left( -\log \sum_{i_d \in \mathcal{I}_d} \exp \left( f(u, i_d) - f(u, i_p) \right) \right) \right]. \tag{13}$$

Review the DPO loss in Eq.(7) and S-DPO loss in Eq.(11). Notably, term $\beta \log \frac{\pi_\theta(e|x_u)}{\pi_{\text{ref}}(e|x_u)}$ is the implicit reward function, denoted by $r_\theta(x_u, e)$ in Section 3.1. According to Section 2, $r_\theta(e, x_u)$ reflects the preference of user $u$ to item $i$ corresponding to title $e$. When the reference model has no knowledge about recommendation, *i.e.,* when $\pi_{\text{ref}}(e|x_u)$ is approximately a uniform distribution, term $r_\theta(x_u, e) = \beta \log \frac{\pi_\theta(e|x_u)}{\pi_{\text{ref}}(e|x_u)}$ exactly reveals absolute preference. Hence, $r_\theta(x_u, e)$ possesses a similar function to $f(u, i)$ .

From this perspective, DPO and S-DPO can be seen as special patterns of BPR and softmax loss, respectively. Given the effectiveness of BPR and InfoNCE loss in recommendation, we argue that sampled-based loss which explicitly compares preferred and dispreferred items such as DPO and S-DPO is more suitable for training LM-based recommenders than only utilizing language modeling loss. Moreover, as softmax loss works better than BPR loss in multi-negative scenarios [20], it can be inferred that S-DPO will be more tailored for multi-negative user preference alignment than DPO.

## 4 Experiments

In this section, we aim to answer the following research questions:

- **RQ1:** How does S-DPO compare with traditional and LM-based sequential recommendation models on performance?
- **RQ2:** How does the LM-based recommender benefit from the multiple negatives in S-DPO?
- **RQ3:** What are the impacts of the essential parameters ($\beta$) on S-DPO?

**Baselines.** We thoroughly compare S-DPO with three categories of recommenders in sequential recommendations: traditional recommenders (GRU4Rec [46], Caser [47], SASRec [48]), LM-enhanced recommenders (MoRec [49]) and LM-based recommenders (LLaMA2 [32], Chat-REC

Table 1: The performance comparison on three real-world datasets. "Rel.Ipv" denotes the relative improvement of S-DPO compared with baselines.

| | | Goodreads | | | LastFM | | | MovieLens | | |
| --- | --- | --- | --- | --- | --- | --- | --- | --- | --- | --- |
| | | HR@1 | ValidRatio | Rel.Ipv | HR@1 | ValidRatio | Rel.Ipv | HR@1 | ValidRatio | Rel.Ipv |
| Traditional | GRU4Rec | 0.3867 | 1.0000 | *70.91%* | 0.2616 | 1.0000 | *153.36%* | 0.3750 | 1.0000 | *40.35%* |
| | Caser | 0.4174 | 1.0000 | *58.34%* | 0.2233 | 1.0000 | *196.82%* | 0.3861 | 1.0000 | *36.31%* |
| | SASRec | 0.3581 | 1.0000 | *84.56%* | 0.2233 | 1.0000 | *196.82%* | 0.3444 | 1.0000 | *52.82%* |
| LM-based | LLaMA2 | 0.0233 | 0.3845 | *2736.48%* | 0.0246 | 0.3443 | *2594.31%* | 0.0421 | 0.4421 | *1150.12%* |
| | ChatRec | 0.3306 | 1.0000 | *99.91%* | 0.3770 | 1.0000 | *75.81%* | 0.2000 | 0.9895 | *163.15%* |
| | MoRec | 0.2877 | 1.0000 | *129.72%* | 0.1652 | 1.0000 | *301.21%* | 0.2822 | 1.0000 | *86.50%* |
| | TALLRec | 0.4983 | 0.9573 | *32.63%* | 0.4180 | 0.9836 | *58.56%* | 0.3895 | 0.9263 | *35.12%* |
| | LLaRA | 0.5292 | 0.9950 | *24.89%* | 0.4508 | 0.9918 | *47.03%* | 0.4737 | 0.9684 | *11.10%* |
| Ours | S-DPO | **0.6609** | 0.9900 | - | **0.6628** | 0.9992 | - | **0.5263** | 0.9895 | - |

[15], TALLRec [14], LLaRA [19]). See detailed introduction and comparison of baselines in Appendix B.

**Datasets.** We conduct extensive experiments on three real-world benchmark datasets which differ in size and domain (Movielens [50], Goodreads[3], and LastFM [51]). Following standard settings of [15, 19], we employ a commonly used metric Hit Ratio@1 (HR@1) for performance evaluation and an additional metric Valid Ratio to evaluate the LM-based methods' ability to generate appropriate responses. See detailed introductions of datasets and evaluation metrics in Appendix B.

**Implementation.** We implement all LM-based recommenders on 4 NVIDIA A100 GPUs. For all LM-based recommenders, we conduct a supervised fine-tuning stage for a maximum of 5 epochs. For S-DPO and its variants, we conduct a preference alignment stage for a further 3 epochs. Different from existing methods, we only optimize loss on item titles and find it effective in recommendation tasks. Refer to Appendix B for more implementation details.

## 4.1 Overall Performance Comparison (RQ1)

Table 1 presents a comparative analysis of the performance of our proposed S-DPO and baselines. Bold and underlined indicate the best and the second-best performance, respectively. We observe that:

- **LM-based recommenders have driven impressive performance breakthroughs compared with traditional recommenders.** Our results reveal that traditional recommenders outperform untuned LM-based recommenders (LLaMA, ChatRec) but fall short compared to LM-based recommenders fine-tuned on historical interactions (TALLRec and LLaRA). It is noted that untuned LM-based recommenders are limited by inadequate instruction-following capabilities (indicated by a low valid ratio) or a lack of domain-specific knowledge (indicated by a suboptimal performance), which highlights the necessity of the supervised fine-tuning stage to further ground the inherent ability of language models down to sequential recommendation tasks. Moreover, MoRec also exhibits suboptimal performance compared to its traditional variant because it leaves the reasoning ability of LM untouched. The superior performance of recent LM-based recommenders indicates the significant roles of knowledge and reasoning ability in language models for recommendation tasks in semantically informative datasets, which highlights the potential of LM-based recommenders.

- **Tailoring language models for recommendation task further boosts the performance of LM-based recommenders.** For LM-based recommenders, the substantial performance gap between fine-tuned and untuned approaches emphasizes the importance of tailoring models for recommendations. TALLRec adapts LM for recommendation by supervised fine-tuning LM on historical interactions, surpassing traditional recommenders. Additionally, LLaRA consistently outperformed TALLRec across all datasets, suggesting that introducing collaborative signals through appropriate item representations is a viable direction for further adapting LM. However, existing LM-based methods adapt LM from either item representation methods or corpus construction, leaving the adaptation of optimization objectives unexplored. Instead, S-DPO aligns the language model with multi-negative

---

[3]https://www.goodreads.com

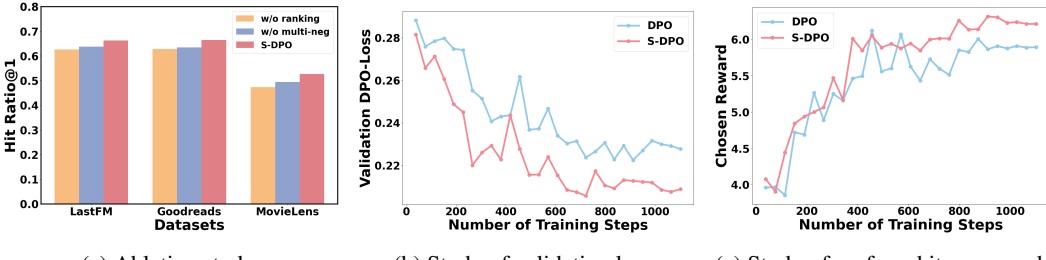

(a) Ablation study.    (b) Study of validation loss.    (c) Study of preferred item reward.

Figure 2: Study on S-DPO. (2a) Ablation study of S-DPO compared with SFT and DPO on three datasets. (2b) Comparison of the trend of validation loss between DPO and S-DPO on LastFM. (2c) Comparison of the reward of preferred items between DPO and S-DPO on LastFM.

user preference data by extending DPO to include a softmax ranking loss, making it a more appropriate loss function for recommendation tasks.

- **S-DPO consistently outperforms all traditional recommenders and the state-of-the-art LM-based recommenders on all datasets.** S-DPO shows an improvement ranging from 11.10% to 47.03% on Hit Ratio@1 compared to the second-best baseline. Building on a supervised fine-tuning stage, we observe a further improvement to the preference alignment stage, which explicitly instills ranking information into LM and utilizes preference data with multiple negative samples. Such superior performance suggests that explicitly tailoring LM for recommendation using user preference data at the training objective level is more effective than other LM-based recommenders. By leveraging the inherent abilities of the LM and incorporating ranking information from user preference data, S-DPO effectively differentiates between preferred and less preferred items. Notably, the preference alignment stage hardly harms the inherent ability of LM, illustrated by a high valid ratio.

## 4.2 Study on S-DPO

**Ablation Study.** To investigate the effect of explicit ranking optimization and multiple negative samples of S-DPO, we compare it with the vanilla supervised fine-tuned model (w/o ranking), and a variant of S-DPO with only a single negative sample (w/o multi-neg), downgrading to pairwise DPO loss. The experimental results are reported in Figure 2a. We can observe that DPO can achieve an overall better performance compared to SFT, which underscores the effectiveness of instilling ranking relationships into existing LM-based recommenders. With a more effective ranking gradient provided by multiple negative samples, S-DPO can further boost performance and achieve the best among all baseline methods and variants.

**Study on the number of negative samples (RQ2).** Benefiting from the utilization of multiple negative pairs in preference data, our S-DPO offers two empirically appealing properties compared to DPO: 1) S-DPO has more effective gradients facilitating the optimization; 2) S-DPO provides a better boost for rewards of preferred items compared to DPO. Figure 2b provides the comparison of validation loss between S-DPO and DPO, illustrating that the loss of S-DPO decreases faster and more significantly. This observation demonstrates that multiple negative pairs provide larger and more meaningful gradients for model optimization, which is attributed to the inherent benefit of S-DPO to mine negative samples [38] (*cf.* Section 3.1).

On the other hand, we study the behavior of S-DPO which is illustrated in Figure 2c. We surprisingly find that S-DPO exhibits continually increasing rewards of preferred items that are more significant and stable than DPO, which shows better effectiveness in distinguishing preferred items and a potential of mitigating data likelihood decline issues[52, 53].

To further verify the superiority of the multiple negative samples of S-DPO compared with DPO, we select the number of negative samples from {1, 3, 5, 8, 10, 15} to conduct experiments to explore the potential of the number of negative samples, with the results depicted in Figure 3a. It can be observed that utilizing multiple negative samples allows the model to achieve better performance than with a single one. Furthermore, as the number of negative samples increases, the model's performance exhibits continual improvements. We attribute this success of S-DPO to more effective ranking

Table 2: Effectiveness comparison between DPO with single negative, a variant of DPO with multiple negatives and S-DPO with the same number of negatives (we set $K$ as 3 to get the performance in this table).

| Datasets | LastFM | | MovieLens | | Goodreads | | Complexity |
|---|---|---|---|---|---|---|---|
| Measure | HitRatio@1 | ValidRatio | HitRatio@1 | ValidRatio | HitRatio@1 | ValidRatio | |
| **DPO-1negative** | 0.6342 | 0.9972 | 0.4947 | 0.9684 | 0.6381 | 0.9900 | $\Theta(2C_{\mathcal{M}}S_t)$ |
| **DPO-$K$negative** | 0.6413 | 0.9964 | 0.4947 | 0.9474 | 0.6628 | 0.9900 | $\Theta(2KC_{\mathcal{M}}S_t)$ |
| **S-DPO-$K$negative** | **0.6477** | 0.9980 | **0.5263** | 0.9895 | **0.6661** | 0.9950 | $\Theta((K+1)(C_{\mathcal{M}}+1)S_t)$ |

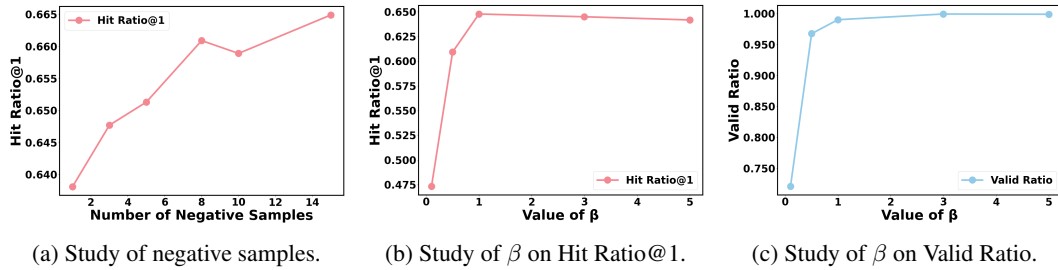

(a) Study of negative samples.   (b) Study of $\beta$ on Hit Ratio@1.   (c) Study of $\beta$ on Valid Ratio.

Figure 3: Studies on values of $\beta$ and negative samples numbers of S-DPO on LastFM. (3a) Performance comparisons with varying numbers of negative samples ($\beta = 1$). (3b) Performance comparisons with varying values of $\beta$ setting negative samples number as 3. (3c) Validity comparisons with varying values of $\beta$ setting negative samples number as 3.

gradients provided by multiple negatives which can be connected to the superior performance of contrastive loss in self-supervised recommendations [38, 39, 54].

To validate the superiority of S-DPO over the DPO variant with multi-negatives, we conduct effectiveness and efficiency comparisons. Table 2 demonstrates that introducing more negative samples benefits both DPO and S-DPO, and S-DPO achieves comparable performance with fewer training steps. We further analyze how S-DPO outperforms DPO in terms of computational efficiency. While the complexity of DPO for $K$ negative samples is $\Theta(2KC_{\mathcal{M}}S_t)$, where $C_{\mathcal{M}} + 1$ denotes the base LLM's computational complexity and $S_t$ represents the size of the training data, S-DPO's complexity for the same number of negatives is reduced to $\Theta((K+1)(C_{\mathcal{M}}+1)S_t)$. This efficiency gain can be expressed by scaling the complexity of DPO by the factor $\frac{1}{2} + \frac{1}{2K} + \frac{1}{2C_{\mathcal{M}}} + \frac{1}{2KC_{\mathcal{M}}}$, highlighting that S-DPO offers significant advantages, especially when working with a larger number of negative samples.

**Study on values of $\beta$ (RQ3).** In S-DPO, $\beta$ is a hyperparameter controlling the deviation of LM from the base reference policy [34]. Typically, a smaller value of $\beta$ implies that the language model is more heavily influenced by the preference signals and vice versa. In this section, we select the value of $\beta$ from {0.1, 0.5, 1, 3, 5} to explore the effect of $\beta$ on S-DPO. As indicated in Figure 3b through 3c, a higher $\beta$ can achieve overall better performance in our task, while a lower $\beta$ may overwhelm the model's learned knowledge from the supervised fine-tuning stage, as evidenced by both low valid ratio and hit ratio. On the other hand, an excessively large $\beta$ prevents the model from effectively learning ranking relationships, leading to suboptimal performance. In all our main experiments and studies, we set $\beta$ as 1 to achieve a balance between ranking signals and inherent knowledge of language models.

## 5   Related Work

### 5.1   LM for Recommendation

Recent advancements in recommendation systems have increasingly incorporated Language Models (LMs) due to their extensive knowledge and robust reasoning abilities. This integration occurs primarily in two forms: LM-enhanced recommenders and LM-based recommenders. LM-enhanced recommenders utilize LM embedding as semantic representations to provide contrastive signals [55–58] or utilize LM as advanced feature extractors improving the representation of user and

item features [59–61]. However, these systems still rely on traditional recommenders for the final recommendation task, which leaves the reasoning ability of LM largely untouched.

On the other hand, LM-based recommenders directly employ LMs for making recommendations. Early works leverage LMs' in-context learning capabilities for zero-shot or few-shot recommendations, demonstrating significant potential [10–12, 15]. However, untuned LM-based recommenders are limited by inadequate instruction-following capabilities and a lack of domain-specific knowledge. To bridge this gap, recent efforts in this category include supervised fine-tuning of LMs on the historical interactions to enhance their performance in recommendation tasks [14, 17, 23, 24, 27]. More recently, researchers have discovered that exploring item representation methods in the finetuning phase may further boost LM's ability for recommendation [30]. This branch of works includes integrating collaborative signals [18, 19, 25, 62–64], adjusting numeric representations [22, 65, 66] or introducing additional item tokens [16, 26, 28, 29].

However, existing finetuned methods follow the training objective of language generation without any specific adjustments for personalized ranking. Different from them, S-DPO proposes to explicitly optimize item ranking information on preference data.

## 5.2 Preference Alignment of Language Models

Reinforcement Learning from Human Feedback (RLHF) [31–33] is a prevalent method of LMs to learn from human preferences. The RLHF pipeline comprises reward model training and reinforcement learning (RL) optimization, the latter of which suffers instability and inefficiency. Direct Preference Optimization (DPO) [34] bypasses the brittle RL phase via a particular reward model parameterization and is thus simpler to implement while still keeping the performance of RLHF.

DPO proves to be effective in many scopes, like NLP [34, 67] and multimodal LMs [35, 68–70]. Besides, several variants have been proposed for further improvement of DPO. $\Psi$PO [71] is a generalization of DPO loss and its representative IPO can better overcome the problem of overfitting. ODPO [36] treats preference pairs differently by stipulating that the likelihood gap of two responses should be greater than a corresponding offset value. KTO [72] utilizes prospect theory for preference alignment tasks. Other variants including GPO [73], $f$-DPO [74], RSO [75] also enhance or expand DPO in various aspects. Despite these contributions, the possibilities for leveraging and further adapting DPO for recommendation are still largely unexplored and few studies discuss extending DPO to handle multi-negative scenarios.

# 6   Limitation

Despite effectiveness, there are several limitations not addressed in this paper. On the one hand, the number of negative samples is capped at 15 in our experiments. The potential of multiple negative samples hasn't been fully explored due to the limited time and computation resources. On the other hand, increasing the number of negative examples inevitably results in higher training costs, a phenomenon that becomes more pronounced as the number of negative examples grows in the context of language models.

# 7   Conclusion

In this work, we devised a principled Softmax-DPO (S-DPO) loss specially tailored for LM-based recommenders, utilizing multiple negatives in preference data to explicitly instill ranking information into LM. Empirically, S-DPO surpasses all baseline models including traditional and LM-based methods on three datasets in sequential recommendation tasks while successfully providing better rewards for preferred items compared to DPO. Grounded by theoretical proof, we bridge S-DPO with the softmax loss in self-supervised recommendations, underscoring the ranking performance of S-DPO and highlighting the critical roles of multiple negatives. Also, we theoretically find that S-DPO has an inherent benefit to mine hard negatives which provide larger and more effective gradients to model optimization, assuring its exceptional capabilities in recommendation tasks. We believe that S-DPO, as a generalization of DPO, provides valuable insights for future LM-based recommenders and has the potential to benefit research fields other than recommender systems[4].

---

[4]The extension to broader impact will be detailed in Appendix D

## Acknowledgments and Disclosure of Funding

This research is supported by the National Science and Technology Major Project (2023ZD0121102), the NExT Research Center, National Natural Science Foundation of China (92270114). This research is also supported by the advanced computing resources provided by the Supercomputing Center of the USTC.

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

## A  Mathematical Derivations

### A.1  Deriving Preference Distribution

The PL model takes the form:

$$p^*(\tau|e_e, i_2, \cdots, e_K, x_u) = \prod_{j=1}^{K} \frac{\exp\left(r(x_u, e_{\tau(j)})\right)}{\Sigma_{l=j}^{K}\exp(r\left(x_u, e_{\tau(l)}\right))}. \tag{14}$$

.

The ranking in multi-negative preference data is $e_p \succ e_d|x_u, \forall e_d \in \mathcal{E}_d$. Our new preference distribution that estimates the probability of the ranking can be derived:

$$\begin{aligned}
p^*(e_p \succ e_d, \forall\, e_d \in \mathcal{I}_d|x_u, e_p, \mathcal{E}_d) &= \sum_{\tau\in\{\tau'|\tau'(1)=p\}} p^*(\tau|x_u, e_p, \mathcal{E}_d)\\
&= \sum_{\tau\in\{\tau'|\tau'(1)=p\}} \prod_{j=1}^{K} \frac{\exp\left(r(x_u, e_{\tau(j)})\right)}{\sum_{l=j}^{K} \exp\left(r(x_u, e_{\tau(l)})\right)}\\
&= \frac{\exp(r(x_u, e_p))}{\sum_{j=1}^{K} \exp(r(x_u, e_j))} \times \sum_{\tau'\in\mathrm{Per}(\mathrm{ind}(\mathcal{I}_d))} \prod_{j=1}^{K-1} \frac{\exp\left(r(x_u, e_{\tau'(j)})\right)}{\sum_{l=j}^{K-1} \exp(r(x_u, e_{\tau'(l)}))}\\
&= \frac{\exp(r(x_u, e_p))}{\sum_{j=1}^{K} \exp(r(x_u, e_j))} \times \sum_{\tau'\in\mathrm{Per}(\mathrm{ind}(\mathcal{I}_d))} p^*(\tau'|x_u, \mathcal{E}_d)\\
&= \frac{\exp(r(x_u, e_p))}{\sum_{j=1}^{K} \exp\left(r(x_u, e_j)\right)},
\end{aligned}$$

wherein $\mathrm{ind}(\mathcal{E}_d)$ denotes the indices of titles in $\mathcal{E}_d$ and $\mathrm{Per}(\mathrm{ind}(\mathcal{E}_d))$ denotes the set of permutations of index set $\mathrm{ind}(\mathcal{E}_d)$. The third equation is because a permutation of $\{1, 2\cdots, K\}$ starting with $p$ can be divided into the prefix $p$ and a subsequent permutation of the rest indices, which is exactly $\mathrm{Per}(\mathrm{ind}(\mathcal{E}_d))$.

### A.2  Connection Between DPO and S-DPO

When $N = 2$, the following equations hold:

$$\mathcal{L}_{\text{S-DPO}}(\pi_\theta; \pi_{\text{ref}}) \tag{Eq.(11)}$$

$$= -\mathbb{E}_{(x_u, e_p, \mathcal{E}_d)}\left[\log\sigma\left(-\log\sum_{e_d\in\mathcal{E}_d}\exp\left(\beta\log\frac{\pi_\theta(e_d|x_u)}{\pi_{\text{ref}}(e_d|x_u)} - \beta\log\frac{\pi_\theta(e_p|x_u)}{\pi_{\text{ref}}(e_p|x_u)}\right)\right)\right]$$

$$= -\mathbb{E}_{(x_u, e_p, e_d)}\left[\log\sigma\left(-\log\exp\left(\beta\log\frac{\pi_\theta(e_d|x_u)}{\pi_{\text{ref}}(e_d|x_u)} - \beta\log\frac{\pi_\theta(e_p|x_u)}{\pi_{\text{ref}}(e_p|x_u)}\right)\right)\right] \quad (N=2)$$

$$= -\mathbb{E}_{(x_u, e_p, e_d)}\left[\log\sigma\left(\beta\log\frac{\pi_\theta(e_p|x_u)}{\pi_{\text{ref}}(e_p|x_u)} - \beta\log\frac{\pi_\theta(e_d|x_u)}{\pi_{\text{ref}}(e_d|x_u)}\right)\right]$$

$$= \mathcal{L}_{\text{DPO}}(\pi_\theta; \pi_{\text{ref}}).$$

Therefore, DPO is a special case of S-DPO.

### A.3  Deriving the Gradient of S-DPO Loss

Let $V(\theta; e_d) = g(e_d, e_p, x_u) = \beta\log\frac{\pi_\theta(e_d|x_u)}{\pi_{\text{ref}}(e_d|x_u)} - \beta\log\frac{\pi_\theta(e_p|x_u)}{\pi_{\text{ref}}(e_p|x_u)}$ and the S-DPO loss takes the following form:

$$\begin{aligned}
&\mathcal{L}_{\text{S-DPO}}(\pi_\theta; \pi_{\text{ref}})\\
&= -\mathbb{E}_{(x_u, e_p, \mathcal{E}_d)}\left[\log\sigma\left(-\log\sum_{e_d\in\mathcal{E}_d}\exp(V(\theta; e_d))\right)\right]
\end{aligned} \tag{15}$$

The gradient of $V(\theta; e_d)$ can be formulated as:

$$\nabla_\theta V(\theta; e_d) = \beta(\nabla_\theta \log \pi_\theta(e_d|x_u) - \nabla_\theta \log \pi_\theta(e_p|x_u)) \tag{16}$$

Using properties of the sigmoid function that $\sigma'(x) = \sigma(x)(1 - \sigma(x)) = \sigma(x)\sigma(-x)$ and thus $((\log \sigma(x))' = \frac{1}{\sigma(x)} \times \sigma(x)\sigma(-x) = \sigma(-x)$, we have:

$$\nabla_\theta \mathcal{L}_{\text{S-DPO}}(\pi_\theta; \pi_{\text{ref}})$$

$$= -\mathbb{E}_{(x_u, e_p, \mathcal{E}_d)} \left[ \nabla_\theta \log \sigma \left( -\log \sum_{e_d \in \mathcal{E}_d} \exp(V(\theta; e_d)) \right) \right]$$

$$= \mathbb{E}_{(x_u, e_p, \mathcal{E}_d)} \left[ \sigma \left( \log \sum_{e_d \in \mathcal{E}_d} \exp(V(\theta; e_d)) \right) \cdot \nabla_\theta \log \sum_{e_d \in \mathcal{E}_d} \exp(V(\theta; e_d)) \right]$$

$$((\log\sigma(x))' = \sigma(-x))$$

$$= \mathbb{E}_{(x_u, e_p, \mathcal{E}_d)} \left[ \sigma \left( \log \sum_{e_d \in \mathcal{E}_d} \exp(V(\theta; e_d)) \right) \cdot \frac{\sum_{e_d \in \mathcal{E}_d} \exp(V(\theta; e_d)) \cdot \nabla_\theta V(\theta; e_d)}{\sum_{e'_d \in \mathcal{E}_d} \exp(V(\theta; e'_d))} \right]$$

$$= -\beta \mathbb{E}_{(x_u, e_p, \mathcal{E}_d)} \left[ \sigma \left( \log \sum_{e_d \in \mathcal{E}_d} \exp(V(\theta; e_d)) \right) \cdot \sum_{e_d \in \mathcal{E}_d} \frac{\nabla_\theta \log \pi_\theta(e_p|x_u) - \nabla_\theta \log \pi_\theta(e_d|x_u)}{\sum_{e'_d \in \mathcal{E}_d} \exp(V(\theta; e'_d) - V(\theta; e_d))} \right]$$

$$(\text{Eq. (16)})$$

$$= -\beta \mathbb{E}_{(x_u, e_p, \mathcal{E}_d)} \left[ \sigma \left( \log \sum_{e_d \in \mathcal{E}_d} \exp(g(e_d, e_p, x_u)) \right) \cdot \sum_{e_d \in \mathcal{E}_d} \frac{\nabla_\theta \log \pi_\theta(e_p|x_u) - \nabla_\theta \log \pi_\theta(e_d|x_u)}{\sum_{e'_d \in \mathcal{E}_d} \exp(g(e'_d, e_d, x_u))} \right]$$

$$(\text{Definition of } V(\theta; e_d))$$

$$= -\beta \mathbb{E}_{(x_u, e_p, \mathcal{E}_d)} \left[ \sigma \left( \log \sum_{e_d \in \mathcal{E}_d} \exp(g(e_d, e_p, x_u)) \right) \cdot \left[ \nabla_\theta \log \pi_\theta(e_p|x_u) - \sum_{e_d \in \mathcal{E}_d} \frac{\nabla_\theta \log \pi_\theta(e_d|x_u)}{\sum_{e'_d \in \mathcal{E}_d} \exp(g(e'_d, e_d, x_u))} \right] \right]$$

The last equation is because:

$$\sum_{e_d \in \mathcal{E}_d} \frac{1}{\sum_{e'_d \in \mathcal{E}_d} \exp(g(e'_d, e_d, x_u))} = \frac{\sum_{e_d \in \mathcal{E}_d} \exp(V(\theta; e_d))}{\sum_{e'_d \in \mathcal{E}_d} \exp(V(\theta; e'_d))} = 1$$

## B  Experimental Settings

### B.1  Baselines

We compare the performance of S-DPO, against both traditional and LM-based baselines to showcase the effectiveness of our method. Specifically, for traditional methods, we have:

- **GRU4Rec** [46] utilizes the GRU (Gated Recurrent Unit) architecture to model sequences, enabling effective prediction in recommendation tasks.
- **Caser** [47] employs both horizontal and vertical convolutional operations to enhance the capture of high-order interactions within item sequences, improving recommendation accuracy.
- **SASRec** [48] incorporates a multi-head self-attention mechanism in its self-attentive sequential recommendation model, facilitating the modeling of intricate sequential data patterns.

For LM-enhanced method, we have:

- **MoRec** [49] advances traditional recommendation systems by incorporating the modality features of items instead of the id feature. we employ BERT for the text encoder and SASRec for the recommendation architecture.

For LM-based methods, we have:

- **LLaMA2** [32] utilized vanilla LLaMA2-7B to directly generate recommendation results through direct prompting.

Table 3: Statistics of datasets.

| Dataset | MovieLens | Goodreads | LastFM |
|---|---|---|---|
| #Sequence | 943 | 6,031 | 1,220 |
| #Items | 1,682 | 4,500 | 4,606 |
| #Interactions | 100,000 | 220,100 | 73,510 |

- **Chat-REC** [15] is implemented based on the framework discussed in [15], we retain user interaction sequences consisting of item titles as use profiles for a fair comparison. We use GPT4 [76] as its primary large language model.
- **TallRec** [14] first propose to transform interaction sequences into textual prompts and then fine-tunes large language models using domain-specific corpus.
- **LLaRA** [19] combines collaborative signals from traditional recommendation systems into the fine-tuning of large language models for improved recommendation performance.

## B.2 Datasets

To evaluate the effectiveness of S-DPO, we conduct experiments on three widely used real-world datasets: Movielens [50], Goodreads[5], and LastFM [51]. The statistics of datasets are illustrated in Table 3. The MovieLens dataset is widely used for movie recommendation tasks and includes user ratings and movie titles, we select the MovieLens100K dataset in our experiment. Similarly, Goodreads is sourced from a social book cataloging website, where users can explore, rate, and review a variety of books. LastFM dataset comprises users' listening history and artists' names from the Last.fm online music platform. Following [19], we maintain their titles as textual descriptions for each dataset. For Goodreads, we remove users and books with less than 20 interactions, which keeps the same as the processing of MovieLens. For all datasets, we organize sequences chronologically before dividing the data into training, validation, and testing sets in an 8:1:1 ratio to prevent any potential information leakage.

## B.3 Implementation Details

We implement all approaches with Python 3.9.7, PyTorch 2.2.2, and transformers 4.38.2 on 4 NVIDIA A100s. We select Llama2-7B [32] as the LM backbone for S-DPO. Following [19], we randomly select prompts from several prompt formats during training and evaluation to ensure flexibility and generality. For optimization of all the traditional methods, the Adam optimizer is employed with a learning rate adjusted to 0.001, and a batch size configured at 256. All models undergo L2 regularization, with coefficients experimentally determined from [1e-3, 1e-4, 1e-5, 1e-6, 1e-7]. In all experiments involving large language models, we train each method for a maximum of 5 epochs using a batch size of 128 and select the checkpoint with the lowest loss on the validation set as the final checkpoint. A warm-up strategy is applied to the learning rate, starting at 5% of its maximum value, and gradually adjusting it through a cosine scheduler throughout the training process. For S-DPO and all of its ablation studies, we further conduct preference training for further 3 epochs with a batch size of 128 and a learning rate of 1e-5. Setting the value of $\beta$ as 1, we search the number of negative samples in [3,5] for the main results. The effects of both factors are further explored in 4.2.

## B.4 Evaluation Metrics

Given that LMs primarily produce textual responses rather than comprehensive item rankings, we utilize a re-ranking metric in line with previous research [19] to assess recommendation performance. For each sequence, a candidate set is constructed by randomly selecting 20 non-interacted items and always includes the correct item. We assess all models based on their ability to pinpoint the correct item within this candidate set, employing the HitRatio@1 (HR@1) metric for performance evaluation. Following [19], we also introduce an additional metric called the Valid Ratio to evaluate the LM-based methods' adherence to instructions and their ability to generate appropriate responses. Due to the difficulty LMs face in producing ranked results for candidate items, position-aware metrics like NDCG are deemed unsuitable for this evaluation.

---

[5]https://www.goodreads.com

Table 4: The performance comparison among three different backbone language models on LastFM and MovieLens.

| | | LLaMA1-7B | | Mistral-7B | | Pythia-2.8B | |
|---|---|---|---|---|---|---|---|
| | | HR@1 | ValidRatio | HR@1 | ValidRatio | HR@1 | ValidRatio |
| LastFM | Vanilla | 0.0465 | 0.5872 | 0.0633 | 0.3648 | 0.0265 | 0.3648 |
| | Language Modeling | 0.5980 | 0.9980 | **0.7828** | 0.9992 | 0.1611 | 0.4281 |
| | DPO | 0.6084 | 0.9976 | 0.7415 | 0.9964 | 0.1896 | 0.4220 |
| | S-DPO (3 negatives) | 0.6285 | 0.9976 | 0.7679 | 0.9972 | 0.1948 | 0.4689 |
| | S-DPO (8 negatives) | **0.6365** | 0.9988 | 0.7820 | 0.9972 | **0.2200** | 0.4685 |
| MovieLens | Vanilla | 0.0316 | 0.5158 | 0.0842 | 0.6737 | 0.0421 | 0.4421 |
| | Language Modeling | 0.3895 | 0.9684 | 0.4211 | 0.9895 | 0.1053 | 0.5684 |
| | DPO | 0.3789 | 0.9684 | 0.4421 | 0.9684 | 0.1271 | 0.8449 |
| | S-DPO (3 negatives) | **0.4526** | 0.9474 | 0.4421 | 0.9895 | 0.1271 | 0.8737 |
| | S-DPO (8 negatives) | **0.4526** | 0.9579 | **0.4947** | 0.9895 | **0.1474** | 0.8737 |

# C  Study on Backbone Language Models

In order to examine whether the superiority of S-DPO loss over traditional language modeling loss can be generalized across different backbone language models, we conducted experiments using models with varying architectures and sizes, including LLAMA1-7b [77], Pythia-2.8b [78], and Mistral-7b [79]. These models were tested on two distinct datasets: LastFM and MovieLens. We evaluated three training approaches: language models fine-tuned with standard language modeling loss, and models further trained using DPO and S-DPO losses. Due to limitations in computational resources and time, we experimented with two variants of S-DPO: S-DPO (3 negatives) and S-DPO (8 negatives). As shown in Table 4, S-DPO consistently outperformed the language modeling loss, enhancing model performance while maintaining or even improving the validity of the generated outputs. Additionally, the performance continued to improve as the number of negative samples increased from 3 to 8.

# D  Broader Impact

We left further exploration of softmax ranking loss in LM including more negative samples and validation on various settings as future works. We believe that S-DPO, a generalization of DPO loss has the potential to benefit other research areas other than recommender systems. This paper presents work whose goal is to advance the field of Machine Learning. There are many potential societal consequences of our work, none of which we feel must be specifically highlighted here.

