# OpenReview forum: "On Softmax Direct Preference Optimization for Recommendation"
_NeurIPS.cc/2024/Conference — NeurIPS 2024 poster_

### Official Review · Reviewer_6tmb · 2024-07-09

**Soundness:** 3
**Presentation:** 3
**Contribution:** 2
**Rating:** 5
**Confidence:** 3

**Summary:**

This paper extends DPO from pairwise (Bradley-Terry) to multi-way comparison (Plackett-Luce), where one positive example is considered better than multiple negative examples. The paper applies this approach in recommendation systems, showing promising results on several standard benchmarks.

From Rafailov et al: “although more general Plackett-Luce ranking models [ 30, 21] are also compatible with the
framework if we have access to several ranked answers”.
This paper shows the derivation of the Plackett-Luce extension. In that sense the technical contribution is somewhat incremental.
I would therefore view this more as an application paper in recommendation systems. However this is not how it is written, most of the focus is on the theoretical analysis extending DPO from Bradley-Terry to Plackett-Luce.

**Strengths:**

* The paper is technically sound and mostly clearly written.
* The experimental results look very promising.

**Weaknesses:**

* The technical contribution seems somewhat incremental.
* Some points need clarification (see questions below).

**Questions:**

1. It is assumed that there is a single positive example which is better than all the negative ones. In reality, multiple items may satisfy users’ interests, so the negative examples should be chosen with care. In the experiments, how are negative samples chosen? This was not clear from the paper. \
Actually, in recommendations it is natural to show a slate of results and the user would click on one (or more), so the unclicked results can serve as negatives. Though this is not the type of data available in benchmarks used here such as MovieLens.

2. In Fig 2a, it seems like most of the gains in performance already exist in SFT. How come this is much better than other fine-tuned models like LLaRA?

3. Fig 2b: the performance of S-DPO seems to still be improving after 1200 iterations, why not run it longer until it stops improving?
Also, since the S-DPO loss and DPO loss are not directly comparable, you should include a figure showing HR@1 as a function of training steps for S-DPO and DPO.

4. Fig 3a: the performance seems to still be increasing, is there a number of samples for which it saturates?

5. MIssing baseline: compare the proposed approach to DPO with pairs consisting of the positive example and each negative example separately (use same negative example choice as S-DPO). This is more computationally expensive but interesting to see how it compares in terms of performance (and runtime).
Related to that, it would also be good to include a comparison of runtimes between S-DPO and DPO (perhaps show the above variant of Fig 2b in terms of runtime, not just of training steps).

Minor:
* L79: empty section
* L132: deviate -> deviating
* L175: the exp should be with the sum
* L180: “faster<space>.”

**Limitations:**

No societal impact concerns.

---

> ### Author Rebuttal · Authors · 2024-08-07
>
> **For Reviewer 6tmb**
>
> We appreciate your comments, some of which inspires us to greatly improve our paper. Below we provide the point-to-point responses to address your concerns and clarify the misunderstandings of our proposed method. If you have additional questions, we would be pleased to discuss them with you.
>
> >**Comment 1: Uncertain contribution**
>
> We acknowledge your statement that original DPO has mentioned the utilization of preference models such as Bradley-Terry and Plackett-Luce to model strict ranking relationships among ranked data samples. We claim that S-DPO is a alternated version of DPO deriving a new preference model from the PL model, which is different from the original PL model and has a softmax structure. The the new preference model focuses on partial ranking relationships among preferred items and dispreferred items, which widely exist in recommendation data.
> Such inherent different in data characteristics makes it nontrivial to properly adapt DPO in recommendation while introducing multiple negatives which is important for recommenders.
> Besides adapting softmax DPO to recommendation data, we also provide theorectical analysis to connect pairwise DPO with BPR loss and connect S-DPO with softmax loss in recommendation such as InfoNCE, ensuring its superior performance. Also, we further analyze the gradient of S-DPO loss and theorectically prove that mining hard negatives is the reason behind S-DPO to provide effective gradient.
>
> LM-based recommenders aim to directly generate the preferred items by first encoding textual prompt then autoregressively decoding, which is different from traditional recommenders which calculate similarity among user and item numerical representations. With such differences, the training losses between LM-based recommenders and conventional recommenders have connections but are strictly under different settings. So traditional loss functions like BPR and InfoNCE are unsuitable for LM-based recommenders.
>
> For LM-based recommenders, introducing ranking information and multiple-negatives has been largely omitted. Besides, there lacks effective methods to introduce multiple negatives into the training pipeline of LM-based recommenders. To our knowledge, we are among the first to point out that multi-negatives is important in LM-based recommenders and propose to effectively instill partial ranking information into LM-based recommenders through an alternated softmax version of DPO.
>
> >**Comment 2: Negtive sampling.**
>
> You are right that negative sampling is a critical area of exploration in recommender systems. In this work, negative samples were randomly selected. We acknowledge the importance of more sophisticated negative sampling strategies, such as popularity-based and similarity-based negative sampling, which we leave as future work.
> The type of data you mentioned, where unclicked results serve as negatives, is indeed common in real-world industrial datasets but is challenging to obtain in benchmark datasets like MovieLens. Within our experimental setting, random sampling serves as a simple yet effective strategy.
>
> >**Comment 3: Gain of performance.**
>
> As line 215-217 mentioned, we optimize loss only on item title and find it effective in recommendation data. For other LM-based baselines, we adopt their official implementation for a fair comparison.
>
> >**Comment 4: Saturation of negative samples.**
>
> We believe that the performance will further improve after more training iterations but as mentioned in limitation section that our computation resourse is limited, so we run all experiments for 3 epochs for fair comparson. We also believe that introducing more negatives may bring more gain but we can only explore part of it due to the same limitation in our computation resources.
>
> >**Comment 5: Efficiency.**
>
> We appreciate your point. To address this, we have now included effectiveness and efficiency comparisons between DPO and S-DPO.
>
> $K$ denote the number of negative items.
>
> $C_\mathcal{M}$ denote the complexity of base model $\mathcal{M}$.
>
> **Time Complexity**
>
> | Methods | S-DPO | DPO |
> |:------:|:------:|:------:|
> | Complexity | $\Theta((K+1)(C_\mathcal{M}+1))$ | $\Theta(2KC\_\mathcal{M}S\_t)$ |
>
>  The complexity of S-DPO scales with the factor $\frac{1}{2}+\frac{1}{2K}+\frac{1}{2C_\mathcal{M}}+\frac{1}{2KC_\mathcal{M}}$ compared to DPO. Since $C\_\mathcal{M}$ is usually large for LLMs, the bigger $K$ is, the smaller the factor is, which means the more efficiency S-DPO will posses compared with DPO.
>
> Empirically, When both considering 3 negative examples and trained on goodreads dataset with 4 A100s, it will take 25h for S-DPO and 53h for DPO, indicating the efficiency of our method.
>
> Additionally, we conducted experiments on three datasets compare DPO with multiple negatives and S-DPO on LLAMA2-7b.
>
> **LastFM**
> | Methods | HR@1 | ValidRatio |
> |:------:|:-------:|:------:|
> | DPO | 0.6342 | 0.9972 |
> | DPO-3neg | 0.6413 | 0.9964 |
> | S-DPO-3neg | **0.6477** | **0.9980** |
>
>
> **MovieLens**
> | Methods | HR@1 | ValidRatio |
> |:------:|:-------:|:------:|
> | DPO | 0.4947 | 0.9684 |
> | DPO-3neg | 0.4947 | 0.9474 |
> | S-DPO-3neg | **0.5263** | **0.9895** |
>
> **Goodreads**
> | Methods | HR@1 | ValidRatio |
> |:------:|:-------:|:------:|
> | DPO | 0.6381 | 0.9900 |
> | DPO-3neg | **0.6661** | 0.9900 |
> | S-DPO-3neg | 0.6628 | **0.9950** |
>
> The results show that S-DPO still achieves leading or comparable performance with less complexity. This can be attributed to S-DPO considering more negative examples during single gradient update, resulting in more effective gradients compared to DPO.

---

> > ### Author Response · Authors · 2024-08-09
> > **Sorry for leading confusion**
> >
> > We apologize for our mistakes. The revised complexity analysis is as follows:
> >
> > $K$ denotes the number of negative items.
> >
> > $C_\mathcal{M}$ denotes the base model $\mathcal{M}$.
> >
> > $S_t$ denotes the size of the S-DPO training set.
> >
> > **Time Complexity**
> >
> > | Methods | S-DPO | DPO |
> > |:------:|:------:|:------:|
> > | Complexity  | $ \Theta((K+1)(C_\mathcal{M}+1)S_t) $  | $\Theta(2KC\_\mathcal{M}S\_t)$ |

---

> > > ### Comment · Reviewer_6tmb · 2024-08-12
> > >
> > > Thanks for clarifying the point about the loss based on partial ranking being different than PL which is based on full ranking. In that case is it not more similar to a multinomial logit choice model than to PL?
> > >
> > > Regarding negative sampling, I understand that natural (non-random) negatives are hard to obtain from public benchmarks. You could potentially run a semi-synthetic setting where the recommender presents a slate of items and the user choice is simulated. For example, you can learn a user utility for each item based on past user ratings and then use the utility as a score in the choice model.
> > >
> > > Thanks for adding experiments with the DPO-neg baseline, I think those should be included in the paper. In 2 out of the 3 settings DPO-neg achieves similar performance -- but with higher computational cost, so you can make that point in the paper.

---

> > > > ### Author Response · Authors · 2024-08-13
> > > > **Response for new questions**
> > > >
> > > > Thanks for your continued interest in our work.
> > > >
> > > > For the first comment. Our loss is based on our proposed preference model $p^*(e_p\succ e_d,\forall e_d\in E_d|x_d)=\frac{{\rm exp}(r(e_p,x_u))}{\sum_{j=1}^K{\rm exp}(r_\theta(e_j,x_u))}$, which is derived in the paper from the PL model. In recommendation scenarios, we follow the RLHF pipeline and finally derive this new preference model which possesses a softmax structure similar to a multinomial logit choice model.
> > > >
> > > > For the second comment,  we agree that it is interesting to explore the effectiveness of natural negatives. Currently, we are working on simulating user choice with LLM to collect simulated behaviors as natural negative samples following your suggestions. The response to this question is on the way, for the necessary time to set up this experiment including training LLM. We will release our results once the experiment is finished.
> > > >
> > > > For the last comment, we appreciate your acknowledgment. All the results of additional experiments will be included in the revised version of the paper.
> > > >
> > > >  If you have additional concerns, we would be more than happy to provide additional clarification. Thank you for your attention.

---

> > > > > ### Author Response · Authors · 2024-08-14
> > > > > **Experimental results for natural negatives**
> > > > >
> > > > > Following your suggestion, we employed GPT-4 to simulate user behaviors and used these simulated user choices as natural negatives. The experimental results are as follows:
> > > > >
> > > > > | Method | HitRatio@1 | ValidRatio |
> > > > > | ------ | ---------- | ---------- |
> > > > > | SDPO-random | 0.6609 | 0.9900 |
> > > > > | SDPO-natural | **0.6689**     | **0.9916**     |
> > > > >
> > > > > The results indicate that incorporating hard negatives is beneficial for S-DPO, highlighting the critical role of negative sampling strategies. We believe that exploring negative sampling is an intriguing direction with significant potential for LM-based recommenders.
> > > > >
> > > > > We appreciate your insightful comments and look forward to further discussion.

---

> > > > > > ### Author Response · Authors · 2024-08-14
> > > > > > **Thank you for your time**
> > > > > >
> > > > > > We are very grateful for your valuable comments and your continual interest in our work. Your input on natural negatives has been of great assistance in refining our approach and leaving space for future exploration. We hope that our responses have addressed most of your concerns. If they have, we would appreciate it if you could consider revising your score. As the rebuttal period is ending soon, we are eager to further discuss any remaining concerns you may have. Is there anything else you’d like to clarify?

---

### Official Review · Reviewer_AMu8 · 2024-07-12

**Soundness:** 2
**Presentation:** 3
**Contribution:** 2
**Rating:** 5
**Confidence:** 4

**Summary:**

proposed a softmax DPO in recommendation domain.

**Strengths:**

1. The paper mainly focus on softmax DPO for recommendation
2. The paper is well wrriten with good experimental validation.
3. The source code is avaliable.

**Weaknesses:**

1. The novely of this work is not high, it seems that they mainly adapt DPO to the recomemndation area, most of the concept are from DPO.

2. what is the complexith of S-DPO, and how can it scale up the large-scale dataset?

**Questions:**

1. The novely of this work is not high, it seems that they mainly adapt DPO to the recomemndation area, most of the concept are from DPO.

2. what is the complexith of S-DPO, and how can it scale up the large-scale dataset?

**Limitations:**

It is unclear how this framework can scale up to large scale dataset.

---

> ### Author Rebuttal · Authors · 2024-08-07
>
> **For Reviewer AMu8**
>
> Thanks for your time and feedback. To address your concerns, we present the point-to-point responses as follows. Looking forward to more discussions with you.
>
> >**Comment 1: Uncertain Novelty.**
>
> Generative LM-based recommenders has recently been explored which aim to directly generate the preferred items by first encoding textual prompt then autoregressively decoding, which is different from traditional recommenders which calculating similarity among user and item numerical representations. With such differences, the training losses between LM-based recommenders and conventional recommenders have connections but they are strictly under different settings.
>
> For LM-based recommenders, introducing ranking information and multiple-negatives has been largely omitted. Besides, there lacks effective methods to introduce multiple negatives into the training pipeline of LM-based recommenders. To our knowledge, we are among the first to point out that multi-negatives is important in LM-based recommenders and propose to effectively instill partial ranking information into LM-based recommenders through an alternated softmax version of DPO.
>
> Adapting DPO to recommendation is a non-trivial task. Conventional DPO utilizes the Bradley-Terry (BT) or the Plackett-Luce (PL) preference model and focus on absolute ranking relationships among data samples, while we also utilize the PL preference model but focus on partial ranking relationships specifically for recommendation data and derive a new preference distribution from the PL model.
> Moreover, we also provide theorectical analysis to connect pairwise DPO with BPR loss and connect S-DPO with softmax loss in recommendation such as InfoNCE. Also, we further analyze the gradient of S-DPO loss and theorectically prove that mining hard negatives is one of the reasons behind S-DPO to provide effective gradient.
>
> >**Comment 2: Uncertain Complexity.**
>
> To address your concern, we have now included analysis  of efficiency comparison between traditional DPO and our S-DPO, focusing on theoretical analysis  and empirical results . This comparison highlights the efficiency gains achieved by S-DPO, supporting its practical value.
>
> Let $K$ denote the number of negative items taken into consideration by S-DPO and $C_\mathcal{M}$ denote the time complexity of the chosen base model $\mathcal{M}$. The time complexity of computing S-DPO loss is $\Theta((K+1)C_\mathcal{M}+K+1)=\Theta((K+1)(C_\mathcal{M}+1))$.
>
> For the DPO loss, the complexity of computing a single positive-negative pair is $\Theta(2C_\mathcal{M})$.
>
> Besides, when an equal number of negative items are added to the DPO training set, the total number of the DPO training set is $K$ times of that of the S-DPO training set. Therefore, let $S_t$ denote the size of the S-DPO training set. The time complexity of training with DPO loss scale to $\Theta(2C_\mathcal{M}\times KS_t)=\Theta(2KC_\mathcal{M}S_t)$.  Although this complexity is in the same magnitude as the time complexity of training with S-DPO loss, the latter scales the complexity of training with DPO loss with the factor $\frac{1}{2}+\frac{1}{2K}+\frac{1}{2C_\mathcal{M}}+\frac{1}{2KC_\mathcal{M}}$.  Since $C\_\mathcal{M}$ is usually large for LLMs, the bigger $K$ is, the smaller the factor is, which means the more efficiency S-DPO will posses compared with DPO.
>
> **Time Complexity**
>
> | Methods | DPO-1neg | S-DPO | DPO |
> |:------:|:-------:|:------:|:------:|
> | Complexity | $\Theta(2C_\mathcal{M}S_t)$ | $\Theta((K+1)(C_\mathcal{M}+1))$ | $\Theta(2KC\_\mathcal{M}S\_t)$ |
>
> > **Comment 3: Large scale Dataset**
>
> We have used large scale benchmark dataset such as goodreads. For extremely large datasets, it might be a common issue for researchers having difficulties to implementation.

---

> > ### Author Response · Authors · 2024-08-09
> > **Sorry for leading confusion about time complexity**
> >
> > We apologize for our mistakes. The revised complexity analysis is as follows:
> >
> > $K$ denotes the number of negative items.
> >
> > $C_\mathcal{M}$ denotes the base model $\mathcal{M}$.
> >
> > $S_t$ denotes the size of the S-DPO training set.
> >
> > **Time Complexity**
> >
> > | Methods | S-DPO | DPO |
> > |:------:|:------:|:------:|
> > | Complexity  | $ \Theta((K+1)(C_\mathcal{M}+1)S_t) $  | $\Theta(2KC\_\mathcal{M}S\_t)$ |

---

> > ### Author Response · Authors · 2024-08-14
> > **Thank you for your time**
> >
> > Thank you very much for your valuable feedback. Your suggestions on complexity comparison have enhanced our method. We hope our replies have resolved most of your concerns. If so, we kindly ask if you might consider revising your score. As the rebuttal period is nearing its end, we would like to discuss any remaining issues you may have. Is there anything else you'd like to discuss?

---

### Official Review · Reviewer_iyxD · 2024-07-13

**Soundness:** 2
**Presentation:** 2
**Contribution:** 3
**Rating:** 5
**Confidence:** 3

**Summary:**

The authors propose a modification to the Direct Preference Optimization (DPO) by incorporating a softmax loss to enhance the training of language model (LM)-based recommender systems.

**Strengths:**

- The paper is well-written.
- The mathematical formulations are clear and appreciated.
- The proposed method is fairly simple yet effective.
- The results are convincing.

**Weaknesses:**

1) Novelty
- The use of multiple negatives in recommendation systems is not new and should be cited. Thus, Research Question 2 (RQ2) lacks novelty as it has already been explored. The novelty of using multiple negatives in LM-based recommender systems is uncertain.

2) Statements
- Equation 8 represents the softmax loss, which has already been applied in other recommender systems like BERT4Rec, whereas Equation 12 does not represent the softmax loss, leading to confusion.
- The claim in lines 225-226 "indicating the significant roles of knowledge and reasoning ability in language models for recommendation tasks." is not provable due to differences in training datasets.
- The statement "However, indicated by the low valid ratio and suboptimal performance, untuned LM-based recommenders are limited by inadequate instruction-following capabilities or a lack of domain-specific knowledge" in lines 228-230 is incorrect as evidenced by high ValidRatio scores of ChatRec.

3) Replicability
- The prompts used in the experiments are not provided, making replication difficult.


4) Evaluation
- NDCG is usually preferred, and metrics like @10 or @20 are more commonly used than @1 in recommendation evaluations. For example, LM-based RecSys works like LLama4Rec use @5.
- In Figure 2a, the statistical significance of the results is not specified. It is also unclear why the results are based on LLama and not LLama2.

5) Comparison
- The comparison of training epochs between LM-based recommenders (5 epochs) and S-DPO (an additional 3 epochs) is unfair.
- In Table 1, the best HR@1 is highlighted in bold, but the same should be done for the ValidRatio column even if your model does not perform the best in that aspect.
- The statistical test used should be specified.
- Optimizing hyperparameters for the proposed model without doing the same for competitor models is unfair. A fair comparison would involve equal time allocation for hyperparameter searches.
- Figure 2b shows that S-DPO decreases faster but remains higher than DPO, making the comparison questionable. Similarly, Figure 2c lacks clarity on whether the values are comparable.
-The statement "On the other hand, an excessively large β prevents the model from effectively learning ranking relationships, leading to suboptimal performance" is not supported by Figure 3b and 3c, which show consistent performance improvement.


6) Related work
- The related work section is more effective when placed after the introduction to better contextualize the state-of-the-art before presenting the new method.
- The discussion on LM for recommendation is not thorough, with many works cited together without detailed descriptions.

**Questions:**

- Consider replacing "side effect" with a more positive synonym.
- The methodology section (2) appears empty; clarify if the true (new) methodology starts at section 3.1.
- Equation (1) needs clarification as i_p and i_d represent items (identifiers) and not scores nor ranks.
- Space before subsection name in line 148
- Line 159: "mutli" --> write "multi"
- Clarify lines 174-180 regarding "hard negative items" and the theoretical basis for S-DPO's ability to mine hard negatives.
- Section 3.2 is unclear, especially the statement "The structural correlation indicates that DPO and S-DPO are more suitable for recommendation than the language-modeling loss."
- The definition of Rel.Ipv should be moved to the caption of Table 1.

**Limitations:**

- The computational effort required for using more negative samples is not considered. It would be beneficial to compute an approximate empirical complexity of the method.
- The use of the Movielens 100K dataset is a limitation due to its small size. A larger dataset like ML-1M would provide better validation while remaining computationally feasible compared to ML-20M.

---

> ### Author Response · Authors · 2024-08-07
>
> **For Reviewer iyxD**
> I would like to express my gratitude for your detailed review and the valuable feedback provided.
>
> >**Comment 1: Uncertain Novelty.**
>
> We agree with you that introducing multiple negatives is important and has been widely explored in conventional recommenders, as we briefly discussed in section 3.2.
> However, as your discovery, LM-based recommenders which has different training paradigm compared with traditional recommenders has just been explored, on which multiple-negatives has largely omitted and lack effectively method to introduce into training pipeline. To our knowledge, we are among the first to point out that multi-negatives is important in LM-based recommenders and propose to effectively instill partial ranking information into LM-based recommenders through an alternated softmax version of DPO.
>
> >**Comment 2: Confusing Statement**
>
> - Clarification of Equation 12:
>   Equation 12 is also an equivalent variant of the softmax loss, which can be rewritten as: $
>   E_{(u,i_p,I_d)}\left[{\rm log}\frac{{\rm exp}(f(u,i_p))}{{\rm exp}(f(u,i_p)) + \sum_{i_d \in \mathcal{I}_d}{\rm exp}(f(u,i_d))}\right] $
>
> - We will modify line 225 to "indicating the significant roles of knowledge and reasoning ability in language models for recommendation tasks in semantically informative datasets."
>
> - Untuned LM-based recommenders achieve suboptimal performance because of inadequate instruction-following capabilities (reflected by a low valid ratio and a low hit ratio) or lack of domain knowledge (reflected by a high valid ratio but a low hit ratio). ChatRec, with GPT-4 as its backend, falls into the second case. We will modify line 228 to avoid confusion.
>
> > **Comment 3: Replicability.**
>
>  In fact, we have uploaded all the code and prompt used in the anonymous link " https://anonymous.4open.science/r/S-DPO-C8E0" which has been attached in the abstract of submitted manuscripts. We will further include all of our prompts in the appendix.
>
> > **Comment 4:  Evaluation.**
>
> - We calculate further HitRatio and NDCG, please refer to supplementary material (Table 2) for results.
>
> - Sorry for leading confusion, the results in Figure 2a is based on LLama2 instead of LLama, which will be modified in the manuscript.
>
> > **Comment 5: Statistical Test.**
>
> We utilize one-sided t-test for all of our statistical test and find the p value less than 0.05 among all shown experiments. Clarification will be included in implementation details.
>
>
> > **Comment 6: Comparison.**
>
> - It has been shown that 5 epochs is enough for model convergence. Follow your suggestions, we conduct SFT for further 3 epochs  and results show that it bring no gain to the final performance.
> - We are adding a dedicated hyperparameter table to provide transparency and demonstrate that each model was given an equal opportunity for optimization. Please refer to supplementary material (Table 1).
> - **Regarding Figure 2b and 2c**:
>   We further ensure that the validation setting for DPO and S-DPO are consistent, allowing for a direct and fair comparison. The revised results are now included in the supplementary material (Figure 1). We compare loss, relative likelihood and absolute likelihood to validate the effectiveness of S-DPO
> - **Regarding Figures 3b and 3c**:
>   A very low β overly prioritizes ranking information, compromising the model's ability to follow instructions, as seen in the decreased valid and hit ratios. Conversely, an excessively high β constrains the model too much by the reference model's standards, leading to lower performance (hit ratio@1). A slight increase in valid ratio with β values greater than 1 suggests better adherence to the reference model's constraints, supporting our interpretation.
>
> >**Comment 7: Related work.**
>
> We agree with your suggestion and we will reorganize our revised version. The lack of in-depth discussion is due to space constraints. We will include a more detailed discussion in the appendix.

---

> ### Author Response · Authors · 2024-08-07
> **We edit the complexity**
>
> >**Comment 8: Unclear section**
>
> - **Synonym.** We will replace it with more positive synonym such as "ancillary benefit" in our manuscript.
> - **Empty section.** Methodology starts at Section 3.1.
> - **Equation (1).** We will modify the notation "$>_u$" to "$\succ_u$" to better distinguish the comparisons between items according to the preferences of user $u$ from the comparisons between scores or ranks.
> - **Lines 174-180.** S-DPO treats gradients of different negative (dispreferred) items differently by assigning the gradient of each negative item with an extra term $\frac{1}{\sum_{e_d \in E_d}{\rm exp}(g(e^\prime_d,e_d,x_u))}$=$\frac{{\rm exp}(r_\theta(x_u,e_d))}{\sum_{e^\prime_d\in E_d}{\rm exp}(r_\theta(x_u,e^\prime_d))}$ . This term reflects the relative reward of each negative item compared with other negative items. We can categorize negative items into two categories: (1) Hard negative items , whose reward $r_\theta(x_u,e_d)$ is relatively high, making it more probable to be chosen by  LM-based recommenders; (2) Easy negative items, whose reward $r_\theta(x_u,e_d)$ is relatively low, making it less likely to be output. For  hard negative items, the extra weight term tends to be larger, leading more decline for likelihood.
> - **Section 3.2.**  "Given effectiveness of BPR and InfoNCE loss in recommendation, we argue that sampled-based loss which explicitly compares preferred and dispreferred items such as DPO and S-DPO is more suitable for training LM-based recommenders than only utilizing language modeling loss."
> - Rest mistakes will be modified in the revised version, thanks for pointing out.
>
>
> > **Comment 9: Complexity.**
>
>
> $K$ denote the number of negative items.
>
> $C_\mathcal{M}$ denote the base model $\mathcal{M}$.
>
> $S_t$ denote the size of the S-DPO training set.
>
> **Time Complexity**
>
> | Methods | S-DPO | DPO |
> |:------:|:------:|:------:|
> | Complexity  | $ \Theta((K+1)(C_\mathcal{M}+1)S_t) $  | $\Theta(2KC\_\mathcal{M}S\_t)$ |

---

> ### Author Response · Authors · 2024-08-14
> **Thank you for your time**
>
> We sincerely appreciate your valuable comments. Your feedback on direct comparisons and complexity analysis is important for improving our method. We hope our responses have satisfactorily addressed most of your concerns. If this is the case, could we kindly ask you to consider adjusting your score? With the rebuttal period coming to a close, we are eager to discuss any further concerns you might have.

---

### Official Review · Reviewer_gGah · 2024-07-15

**Soundness:** 3
**Presentation:** 2
**Contribution:** 2
**Rating:** 4
**Confidence:** 4

**Summary:**

Inspired by advancements in Direct Preference Optimization (DPO) and the success of softmax loss in recommendations, the paper propose Softmax-DPO (S-DPO). S-DPO enhances LM-based sequential recommenders by incorporating multiple negative samples in preference data.The paper demonstrate the superiority of S-DPO in modeling user preferences and boosting recommendation effectiveness.

**Strengths:**

1. DPO, a method that considers preferences in NLP, has been extended to LM-based recommender systems, expanding NLP techniques to the recommender system domain.

2. Building upon related previous work such as reinforcement learning from human feedback (RLHF) and DPO, the motivation and approach of this paper are well-founded.

**Weaknesses:**

In the recommender system domain, considering preferences in the loss function is not particularly new. However, experiments and explanations for this are lacking.

1. There are no experiments comparing S-DPO with other preference-based losses in recommender systems (BPR, InfoNCE, as the paper mentioned ). If such comparisons are not feasible, the reasons for this are insufficiently explained.

2. The experiments lacked the application of S-DPO loss to various base models. Consequently, the general applicability of S-DPO was not demonstrated.

3. Running DPO multiple times and S-DPO appear conceptually similar. Therefore, there are no experiments comparing their efficiency. A comparison of time or memory usage seems necessary.

**Questions:**

Same as weaknesses

Is there a justification for not conducting the experiment described above?

**Limitations:**

LM's methodology was brought to the Recommender system. However, there seems to be a lack of discussion in the Recommender system.

BPR loss is typically not used in sequential scenarios, so it would have been beneficial to emphasize that S-DPO specifically addresses preferences in sequential contexts. Additionally, conducting more experiments to validate this aspect would have been advantageous.

---

> ### Author Rebuttal · Authors · 2024-08-07
>
> **For Reviewer gGah**
>
> Your main suggestions about considering additional base models help us substantiate wide applicability of S-DPO.
>
> >**Comment1: Lack comparison with traditional preference-based losses.**
>
> Thank you for your insightful question, which raises a profound and previously unexplored issue: whether traditional preference-based losses can be directly applied to LM-based recommenders.
>
> Following your suggestion, we attempted to train LM-based recommenders using BPR loss and InfoNCE loss.
> However, due to the significant differences between losses designed for language models (e.g., SFT, DPO) and discriminative preference-based losses, and given the time constraints of the rebuttal period, we have not yet obtained results within a reasonable range. We will continue to adjust parameters and training methods over the next discussion week to provide a more reliable conclusion.
> We greatly appreciate your question, looking forward to more discussions with you.
>
> Additionally, we want to emphasize that we have compared preference-based losses in traditional recommenders, such as GRU4Rec-BPR.
>
>
>
> > **Comment 2: General applicability of S-DPO**
>
> Great point! Following your suggestion, we have **extended our experiments** to include more base models.
>
> We selected language models with different architectures (**LLAMA1-7b, Pythia-2.8b, Mistral-7b**), on varying datasets (**LastFM, MovieLens**), and of different sizes to perform experiments.
> We compared untuned LMs, LMs with only SFT as the training loss, and LMs with DPO and S-DPO for further training.
> Due to computational resource and time limitations, we experimented with S-DPO using **3 and 8** negative samples on two datasets.
>
>
> **Results on LastFM**
>
> **LLAMA1-7b**
> | Methods | HR@1 | ValidRatio |
> |:------:|:-------:|:------:|
> | LLAMA1 | 0.0465 | 0.5872 |
> | SFT | 0.5980 | <u>0.9980<u> |
> | DPO | 0.6084 | 0.9976  |
> | S-DPO-3neg | <u>0.6285</u> | 0.9976  |
> | S-DPO-8neg | **0.6365** | **0.9988** |
>
> **Pythia-2.8b**
> | Methods | HR@1 | ValidRatio |
> |:------:|:-------:|:------:|
> | Pythia | 0.0265 | 0.3648 |
> | SFT | 0.1611 | 0.4281 |
> | DPO | 0.1896 | 0.4220 |
> | S-DPO-3neg | <u>0.1948</u> | **0.4689** |
> | S-DPO-8neg | **0.2200** | <u>0.4685</u> |
>
> **Mistral-7b**
> | Methods | HR@1 | ValidRatio |
> |:------:|:-------:|:------:|
> | Mistral | 0.0633 | 0.7475|
> | SFT | **0.7828** | **0.9992** |  |
> | DPO | 0.7415 | 0.9964 |  |
> | S-DPO-3neg | 0.7679 | <u>0.9972</u> |  |
> | S-DPO-8neg | <u>0.7820</u> | <u>0.9972</u> |  |
>
> **Results on MovieLens**
>
> **LLAMA1-7b**
>
> | Methods | HR@1 | ValidRatio |
> |:------:|:-------:|:------:|
> | LLAMA1 | 0.0316 | 0.5158 |
> | SFT | 0.3895 | **0.9684** |
> | DPO | 0.3789 | **0.9684** |
> | S-DPO-3neg | **0.4526**  | 0.9474 |
> | S-DPO-8neg | **0.4526** | 0.9579 |
>
> **Pythia-2.8b**
> | Methods | HR@1 | ValidRatio |
> |:------:|:-------:|:------:|
> | Pythia | 0.0421 | 0.5895 |
> | SFT | 0.1053 | 0.5684 |
> | DPO | <u>0.1271</u> | <u>0.8449</u> |
> | S-DPO-8neg | **0.1474** | **0.8737** |
>
> **Mistral-7b**
> | Methods | HR@1 | ValidRatio |
> |:------:|:-------:|:------:|
> | Mistral | 0.0842 | 0.6737 |
> | SFT | 0.4211 | 0.9894 |
> | DPO | <u>0.4421</u> | 0.9684 |
> | S-DPO-3neg | <u>0.4421</u> | **0.9895** |
> | S-DPO-8neg | **0.4947** | **0.9895** |
>
> The experimental results indicate that DPO generally enhances the performance of SFT across different language models, demonstrating the importance of preference information for recommendation tasks.
> By incorporating multiple negative examples, S-DPO can achieve further performance improvements on top of DPO.
> Moreover, as the number of negative examples increases, the model's performance also improves.
>
>
>
>
>
> >**Comment 3: Comparison between DPO and S-DPO.**
>
> We appreciate your valuable comment.
> To address this, we have **added effectiveness and efficiency comparisons between DPO and S-DPO**.
>
> $K$: the number of negative items.
>
> $C_\mathcal{M}$: the complexity of base model $\mathcal{M}$.
>
> **Time Complexity**
> | Methods | S-DPO | DPO |
> |:------:|:------:|:------:|
> | Complexity | $O((K+1)(C_\mathcal{M}+1))$ | $O(2K C_\mathcal{M} S\_t)$ |
>
>  The complexity of S-DPO scales with the factor $\frac{1}{2}+\frac{1}{2K}+\frac{1}{2C_\mathcal{M}}+\frac{1}{2KC_\mathcal{M}}$ compared to DPO. Since $C_\mathcal{M}$ is usually large for LLMs, the bigger $K$ is, the smaller the factor is, which means more efficiency S-DPO will posses compared with DPO.
>
> Empirically, When both considering 3 negative examples and trained on goodreads dataset with 4 A100s, it will take 25h for S-DPO and 53h for DPO, indicating the efficiency of our method.
>
> Additionally, we conducted experiments on three datasets compare DPO with multiple negatives and S-DPO on LLAMA2-7b.
>
> **LastFM**
> | Methods | HR@1 | ValidRatio |
> |:------:|:-------:|:------:|
> | DPO | 0.6342 | 0.9972 |
> | DPO-3neg | 0.6413 | 0.9964 |
> | S-DPO-3neg | **0.6477** | **0.9980** |
>
>
> **MovieLens**
> | Methods | HR@1 | ValidRatio |
> |:------:|:-------:|:------:|
> | DPO | 0.4947 | 0.9684 |
> | DPO-3neg | 0.4947 | 0.9474 |
> | S-DPO-3neg | **0.5263** | **0.9895** |
>
> **Goodreads**
> | Methods | HR@1 | ValidRatio |
> |:------:|:-------:|:------:|
> | DPO | 0.6381 | 0.9900 |
> | DPO-3neg | **0.6661** | 0.9900 |
> | S-DPO-3neg | 0.6628 | **0.9950** |
>
> The results show that S-DPO still achieves leading or comparable performance with less complexity. This can be attributed to S-DPO considering more negative examples during single gradient update, resulting in more effective gradients compared to DPO.
>
> > **Limitation 1: Lack of discussion in the recommender system.**
>
> We are not sure we fully understand your point. We discussed the LM-based recommender and traditional recommender in related works.
>
> > **Limitation 2: conducting more experiments and emphasizing S-DPO specifically addresses preferences in sequential recommendation**
>
> Thanks for your valuable comments! Following your suggestion, we added more experiments and will emphasize the important role of S-DPO in sequential recommendation.

---

> > ### Author Response · Authors · 2024-08-09
> > **Sorry for leading confusion**
> >
> > We apologize for our mistakes. The revised complexity analysis is as follows:
> >
> > $K$ denotes the number of negative items.
> >
> > $C_\mathcal{M}$ denotes the base model $\mathcal{M}$.
> >
> > $S_t$ denotes the size of the S-DPO training set.
> >
> > **Time Complexity**
> >
> > | Methods | S-DPO | DPO |
> > |:------:|:------:|:------:|
> > | Complexity  | $ \Theta((K+1)(C_\mathcal{M}+1)S_t) $  | $\Theta(2KC\_\mathcal{M}S\_t)$ |

---

> > ### Author Response · Authors · 2024-08-14
> > **Experimental results on comparing S-DPO with other preference-based losses (BPR and InfoNCe)**
> >
> > It is noted that traditional similarity-based training losses are not well-suited for generative language model-based recommenders, and thus cannot be directly applied in our context. To address this, we adapted BPR and InfoNCE losses within the language model training framework, employing random negative sampling on the LastFM dataset using Llama2-7b. The experimental results are summarized below:
> >
> > | Method     | HitRatio@1 | ValidRatio |
> > | ---------- | ---------- | ---------- |
> > | LLAMA2     | 0.0233     | 0.3854     |
> > | LLAMA2-BPR     | 0.0008     | 0.0152     |
> > | LLAMA2-InfoNCE | 0.0029     | 0.0246     |
> > | LLAMA2-SDPO     | **0.6609**     | **0.9900**     |
> >
> > From these results, we observe that adapting traditional losses for LM-based recommenders significantly diminishes the inherent capabilities of language models, while also failing to effectively capture domain-specific knowledge. This leads to a marked decrease in both hit ratio and valid ratio.
> >
> > From this experiment, we can see that traditional recommendation losses cannot be directly adapted to LM-based recommenders. Therefore, incorporating explicit ranking information and multiple negative examples into LM-based recommenders is a nontrivial task. S-DPO addresses this challenge by introducing a partial order relationship into PL preference modeling and further deriving the effective loss with softmax structure.
> >
> > We look forward to further discussing these findings with you.

---

> > ### Author Response · Authors · 2024-08-14
> > **Thank you for your time**
> >
> > We greatly appreciate your valuable comments. Your insights on various base models and comparisons between DPO and S-DPO have significantly helped us improve our paper. We hope our responses have addressed most of your concerns. If they have, would you consider raising your score? If you have any remaining concerns, we would be eager to discuss them further, especially as the rebuttal period is about to end. Do you have any additional questions?

---

### Author Rebuttal · Authors · 2024-08-07

We are delighted to see the contributions of our paper have been acknowledged by the majority of the Reviewers. Specifically, we appreciate the Reviewers' recognition of our clarity in presentation (iyxD, AMu8, 6tmb), well-founded theoretical analysis (gGah, iyxD, 6tmb) and effectiveness (iyxD, AMu8, 6tmb).

We appreciate all the reviewers for their valuable comments and suggestions. This helped improve our submission and better strength our claims. Taking into account suggestions of Reviewers, we have summarized the updates to the paper as follows:

- **Experiments on three language model backbones compared S-DPO with DPO and SFT.**  Following the suggestions of Reviewer gGah, we have incorporated three additional language model backbones to validate the generalization ability of S-DPO.
- **Comparison experiments of effectiveness and efficiency analysis of DPO and S-DPO under the same negative samples.** Addressing the concerns raised by Reviewers gGah, iyxD, AMu8 and 6tmb, we have conducted experiments to validate the effectiveness of S-DPO with better efficiency compared with DPO.
- **More detailed explanations.** In response to Reviewers gGah, iyxD, AMu8 and 6tmb, we provide detailed explanation to clarify some of our statements, address questions and provide more understanding of our method.
- **Details about S-DPO.** We have incorporated detailed discussion about S-DPO, including hyperparameters selection, data likelihood decline issue and gradient analysis, to address the concerns of Reviewers iyxD and 6tmb.

We have tried our best to address the main concerns raised by reviewers in limited time and we hope that these improvements will be taken into consideration. We also present the point-to-point responses for each reviewer below.

---

### Decision · Program_Chairs · 2024-09-25

**Decision:**

Accept (poster)

**Comment:**

The paper extends DPO to Softmax-DPO for language model based recommendation for user preference alignment inspired by the success of softmax loss in recommendation. While reviewers expressed concerns on novelty, they appreciate the simplicity of the method and solid results, and agree the reasons to accept outweighs reasons to reject after rebuttal.